# Study of Magnetic Properties of Fe_100-x_Ni_x_ Nanostructures Using the Mössbauer Spectroscopy Method

**DOI:** 10.3390/nano9050757

**Published:** 2019-05-17

**Authors:** Kayrat K. Kadyrzhanov, Vyacheslav S. Rusakov, Maxim S. Fadeev, Tatyana Yu. Kiseleva, Artem L. Kozlovskiy, Inesh E. Kenzhina, Maxim V. Zdorovets

**Affiliations:** 1Engineering Profile Laboratory, L.N. Gumilyov Eurasian National University, Astana 010008, Kazakhstan; kayrat.kadyrzhanov@mail.ru (K.K.K.); kenzhina@physics.kz (I.E.K.); mzdorovets@inp.kz (M.V.Z.); 2Department of General Physics, M.V. Lomonosov Moscow State University, Moscow 119991, Russia; rusakov@phys.msu.ru (V.S.R.); maxim.fadeev.msu@mail.ru (M.S.F.); kiseleva_tatiana@mail.ru (T.Y.K.); 3Laboratory of Solid State Physics, The Institute of Nuclear Physics, Almaty 050032, Kazakhstan; 4Kazakh-Russian International University, Aktobe 030006, Kazakhstan; 5Department of Intelligent Information Technologies, Ural Federal University, Yekaterinburg 620075, Russia

**Keywords:** magnetic nanostructures, iron–nickel, magnetic properties, Mössbauer spectroscopy

## Abstract

Hyperfine interactions of ^57^Fe nuclei in Fe_100-x_Ni_x_ nanostructures synthesized in polymer ion-track membranes were studied by Mössbauer spectroscopy. The main part of obtained nanostructures was Fe_100-x_Ni_x_ nanotubes with bcc structure for 0 ≤ x ≤ 40, and with fcc structure for 50 ≤ x ≤ 90. The length, outside diameter and wall thickness of nanotubes were 12 μm, 400 ± 10 nm and 120 ± 5 nm respectively. For the studied nanotubes a magnetic texture is observedalong their axis. The average value of the angle between the direction of the Fe atom magnetic moment and the nanotubes axis decreases with increasing of Ni concentration for nanotubes with bcc structure from ~50° to ~40°, and with fcc structure from ~55° to ~46°. The concentration dependences of the hyperfine parameters of nanotubes Mössbauer spectra are qualitatively consistent with the data for bulk polycrystalline samples. With Ni concentration increasing the average value of the hyperfine magnetic field increases from ~328 kOe to ~335 kOe for the bcc structure and drops to ~303 kOe in the transition to the fcc structure and then decreases to ~290 kOe at x = 90. Replacing the Fe atom with the Ni atom in the nearest environment of Fe atom within nanotubes with bcc structure lead to an increase in the hyperfine magnetic field by “6–9 kOe”, and in tubes with fcc structure—to a decrease in the hyperfine magnetic field by “11–16 kOe”. The changes of the quadrupole shift and hyperfine magnetic field are linearly correlated with the coefficient −(15 ± 5)·10^−4^ mm/s/kOe.

## 1. Introduction

Over the past few decades, nanotechnology has become one of strategic directions of industrial and industrial development. The increased interest in nanostructures is based on the real possibility of their practical application in various fields of science and technology, ranging from microelectronics, catalysts, magnetic media, to alternative energy sources and targeted drug delivery [1,2,3,4,5]. Such interest is due to the unique properties of nanostructures, as well as the possibility of obtaining nanostructures with different shapes and geometries: wires, dendrites, cubes, tubes, spheres, etc. [4,5,6,7,8]. Moreover, in most cases, it is the form that determines both the structural characteristics, and the field of nanostructures application. Today, an enormous amount of work has been done on methods of obtaining and synthesizing nanostructures, as well as evaluating the effect of various parameters on the physical-chemical, structural, conducting, and magnetic properties [7,8,9,10,11,12,13,14,15]. More than a dozen methods have been developed for producing nanostructures of one, two, and more component structures [13,14,15,16,17], layered nanowires in which layers of different thicknesses [18,19] are used in microelectronics, and Core-Shell-type structures [20,21,22], which have proven to be promising targets for biomedical applications, etc. Among the variety of nanostructure forms, hollow magnetic nanostructures in the form of tubes [23,24,25] are of the greatest interest. Increased interest in this form is due from a fundamental point of view: that is associated with the miniaturization of dimensions and structural and magnetic properties, and with the wide possibilities of practical application of nanotubes; it is also due to the fact that many properties, particularly the magnetic texture and orientation of magnetic domains, are due not only to the phase composition, but also to the geometric characteristics of the structure [26,27]. Among the variety of different compositions of nanostructures, iron-containing (Fe_100-x_Ni_x_) nanostructures are incredibly interesting due to their magnetic characteristics and have found wide application in catalysis, magnetic carriers with ultra-high recording density, and biomedicine [28,29,30,31,32]. Despite the large number of papers devoted to the study of Fe_100-x_Ni_x_ nanostructures properties, interest in fundamentals of the influence of various factors on magnetic properties and ultra-fine magnetic texture has not faded [33,34,35,36]. The possibility of controlling the magnetic and structural characteristics, studying the effect of the phase composition on the physical-chemical properties gives grounds for further research in this area [37,38,39,40,41,42,43].

The paper presents the results of a systematic study of the influence of the phase composition and nickel concentration in the structure on magnetic properties of Fe_100-x_Ni_x_ nanotubes. The method of electrochemical synthesis into the pores of polymer templates was used as a method of obtaining nanostructures in tube form. X-ray phase and energy dispersive analysis were used to study phase and elemental composition. The evaluation of the influence of phase composition on magnetic properties was studied by Mössbauer spectroscopy method that allows studying the domain structure and hyperfine parameters of the magnetic field, as well as the orientation of magnetic domains.

## 2. Materials and Methods

### 2.1. Obtaining Fe–Ni Nanostructures

The scheme for obtaining hollow Fe_100-x_Ni_x_ nanostructures using the template synthesis method is presented in Figure 1. The first stage was preparation of PET-based templates (Hostaphan trend, Mitsubishi Polyester Film Corp., Wiesbaden, Germany), which were irradiated at DC-60 heavy ion accelerator with specified parameters. After irradiation, the polymer films were etched in a solution of sodium hydroxide (2.2 M) at a temperature of 85 ± 2 °C for 210 s. These conditions allowed the obtainment of cylindrical pores with a diameter of 380 ± 10 nm (according to SEM images, Figure 1b).

The second stage was the electrochemical synthesis of hollow nanostructures into the pores of polymer templates. To obtain cylindrical nanostructures in the form of tubes a thin layer of gold, no more than 25–30 nm thick, was deposited on one side of the polymer template, which was the contact layer for the start of nucleation processes along the pore walls. The composition of electrolyte solution used to obtain Fe and Fe_100-x_Ni_x_ nanostructures was the following: FeSO_4_ × 7H_2_O, NiSO_4_ × 7H_2_O in the required molar ratio, boric and ascorbic acids. All used chemical reagents had purity of reagent grade (content of the main component is above 98%) or analytical grade (content of the main component is more than 99%). The difference of the applied potentials was 1.75 V. The electrolyte temperature was 25 ± 2 °C. The choice of these synthesis conditions allowed us to control with high accuracy the geometry of resulting nanostructures, as well as the uniformity of composition over the entire length of the nanotubes. The growth of nanostructures was monitored by chronoamperometry using an Agilent 34410A multimeter (Agilent Technologies, Santa Clara, CA, USA).

### 2.2. The Study of Structural Characteristics and Elemental Composition

The structural characteristics of nanotubes were studied using a scanning electron microscope «Hitachi TM3030» (Hitachi Ltd, Chiyoda, Tokyo, Japan). The elemental composition was studied using the system EDA Bruker Flash MAN SVE (Bruker, Karlsruhe, Germany) at an accelerating voltage of 15 kV. The deviation from the expected concentrations of Ni and Fe in the synthesis process did not exceed 2%. Table 1 presents the data of energy dispersive analysis of investigated samples.

### 2.3. Methods of Characterization

The phase composition and crystal structure of the samples were determined on an Empyrean Panalytical diffractometer (Malvern Panalytical B.V., Eindhoven, The Netherlands) (CuKα, λ = 1.5405 Å) in the Bragg-Brentano geometry at a voltage of U = 40 kV and an amperage of I = 40 mA. The program HighScore Plus and the international database ICDD PDF4 were used. The main part of the Fe–Ni nanostructures was Fe_100-x_Ni_x_ nanotubes with a body-centered cubic (bcc) structure (space group Im3m) for 0 ≤ x ≤ 40 and with a face-centered cubic (fcc) structure (space group Fm3m) for 50 ≤ x ≤ 90. Figure 2a shows, as an example, the diffractograms of some of the studied samples in which the nickel concentration was lower than the iron concentration. It can be seen that the intensity peaks in these diffractograms are at an angle of 2*ϑ*, equal to 44.6°, 65.0° and 82.3°, with the corresponding Miller indices: (011), (002) and (112), which confirmed the body-centered cubic structure (bcc). Figure 2b shows the diffraction patterns for samples in which the nickel content exceeds the iron content. Similarly, from the obtained data on diffraction angles of 2*ϑ*, the values of which were 44.0°, 51.0° and 75.3° with the corresponding Miller indices (111), (002) and (022), the crystal structure was determined. For these samples, a face-centered cubic structure (fcc) was observed. The diffraction line at an angle of ~53.7°, observed for all samples, corresponds to the PET pattern template. Figure 2c shows a plot of the change in parameters of the crystal lattice on nickel concentration in the structure in comparison with the literature data.

According to the obtained data, the variation of the crystal lattice parameter from the nickel concentration revealed that for the bcc structure there is a slight increase in the unit cell parameter, and for fcc there is a decrease. Note that the phase diagram of synthesized nanostructures is in good agreement with the phase diagram for bulk samples obtained in [41].

The Mössbauer studies were carried out on a MS1104Em spectrometer (Chernogolovka, Russia) in the constant acceleration mode with a triangular change in the Doppler velocity of the source movement relative to the absorber. All spectra were measured at room temperature. The ^57^Co nuclei in a Rh matrix served as the source. The Mössbauer spectra were processed in SpectrRelax software using techniques based on the reconstruction of hyperfine parameter distribution and model fitting of spectra taking the a priori information about the subject of the study into account [45,46].

All spectra of the Fe_100-x_Ni_x_ nanotubes were also processed as part of the model interpretation. In the model used, the following assumptions were made. First, a random distribution of Ni atoms over the positions of Fe atoms was assumed, i.e., obeying the binomial probability distribution:
(1)$$P_{n}(m)=\frac{n!}{m!(n-m)!}x^{m}{\left(1-x\right)}^{n-m}$$
where *n* is the number of atoms in the nearest environment of the Fe atom, *m* = 0, 1, ..., *n* is the number of Ni atoms in the nearest environment of the Fe atom, *x* = *x*/100% is the concentration of Ni atoms. In the case of the bcc structure, *n* = eight atoms located in the nearest environment of the Fe atom, and in the case of the fcc structure, *n* = 12 atoms.

Taking into account the binomial distribution of Ni atoms in Fe_100-x_Ni_x_ nanotubes, the Mössbauer spectra were processed using a model consisting of a superposition of two quadrupole doublets, a Zeeman sextet, a corresponding iron-containing magnetically ordered impurity, and sextets corresponding to nanotubes (nine sextets for bcc structure and 13 sextets for fcc structure).

Secondly, within the model used, the intensity ratios of Zeeman sextets corresponding to different environments of Fe atoms in nanotubes were assumed to be equal to probability ratios according to the binomial distribution for different concentrations of Ni atoms:*I_1_*:*I_2_*:…:*I_m_*:…:*I_n_* = *P_n_*(0):*P_n_*(1):…:*P_n_*(*m*):…:*P_n_*(*n*)(2)

Third, an additive dependence of the hyperfine parameters *H_n_*, *δ*, and *ε* on the number of m Ni atoms in the nearest environment of the Fe atom was assumed:
(3)$$H_{n}(m)=H_{n}(0)+m\Delta H_{n},$$
(4)δ(m)=δ(0)+mΔδ,
(5)ε(m)=ε(0)+mΔε.


When deciphering the Mössbauer spectra the following were varied: the width of the resonance lines Γ, changes in the hyperfine magnetic field Δ*H_n_*, the shift of the Mössbauer line Δ*δ* and quadrupole displacement Δ*ɛ*, caused by replacing the Fe atom by the Ni atom in the nearest atomic environment of the Fe atom, the hyperfine parameters *H_n_*(0), *δ*(0) and *ɛ*(0), corresponding to the absence of Ni atoms in the nearest environment of the Fe atom. In this case, the value of the parameter α, which defines the shape of the resonance line, was fixed to zero, i.e., the description of resonance lines shape was carried out using the Lorentz function.

## 3. Results

### 3.1. Results of Restoring the Distribution of Ultrathin Parameters of the Mössbauer Spectra

All Mössbauer spectra were obtained at room temperature and in the general case they consist of pairs of Zeeman sextets and quadrupole doublets. The main relative contribution to the intensity is made by a sextet with broadened lines, corresponding to Fe_100-x_Ni_x_ nanotubes. The second sextet corresponds to a magnetically ordered iron-containing oxide impurity. Its contribution did not exceed 4%, and the values of the hyperfine magnetic field of this sextet were in the range of “16–489 kOe. Whereas quadrupole doublets could be attributed to Fe^3+^ cations in iron-oxide formed during the synthesis of nanotubes.

Figure 3 shows the Mössbauer spectrum of Fe nanotubes (left) and the result of reconstructing the distribution of the hyperfine magnetic field (right). It can be seen that the distribution maximum is reached at ~328 kOe. The shifts of the Mössbauer line *δ* and the quadrupole shift *ε* are practically zero, which corresponds to the structure of α-Fe. As a result of processing the spectrum within the restoration of ultrathin parameters, the angle *ϑ* between the magnetic moment of the Fe atom and the nanotube axis was determined as (47 ± 1°). It indicates the presence of some magnetic texture along the axis of the nanotubes.

Figure 4, Figure 5 and Figure 6 show the Mössbauer spectra of Fe_100-x_Ni_x_ nanotubes with concentrations of Ni x = 10–30%, 40–60%, 70–90%, respectively.

As we can see, for Fe_100-x_Ni_x_ nanotubes, the broadening of the hyperfine magnetic field distributions was noticeable compared with Fe nanotubes (Figure 4). With an increase in nickel concentration in the case of 10 ≤ x ≤ 40, the distribution maximum was observed in the direction of increasing hyperfine magnetic field (to the right), and at 50 ≤ x ≤ 90 in the direction of decreasing hyperfine magnetic field (to the left).

As a result of the analysis of the recovered distributions, the average values of the hyperfine parameters of the Mössbauer spectra were obtained and their concentration dependences were plotted. Figure 7a shows the concentration dependence of the average value ${\bar{H}}_{n}$ of the hyperfine magnetic field *H_n_* on ^57^Fe nuclei. It can be seen that this dependence consisted of two regions, the transition between which was accompanied by a jump in ${\bar{H}}_{n}$. On the left, for the bcc structure, there was an increase in the average values of the hyperfine magnetic field from ~328 kOe to ~335 kOe, and on the right, for the fcc structure, a decrease from ~303 kOe to ~290 kOe. Table 2 presents the data for hyperfine parameters for all samples.

The dependence character qualitatively coincides with the behavior of literature data for massive polycrystalline Fe_100-x_Ni_x_ samples, denoted by hollow polygons (Figure 7a).

Concentration dependences for the mean shifts of the Mössbauer line *δ* and quadrupole shift *ε*, shown in Figure 7b,c, respectively, were also obtained. The mean shift of the Mössbauer line increased from about zero to ~0.045 mm/s at 0 ≤ x ≤ 40 and then decreased to ~0.02 mm/s at 50 ≤ x ≤ 90. The average values of the quadrupole shift were close to zero (ε < 0.01 mm/s) for all samples, but negative for the bcc structure and positive for the fcc structure. The behavior of the obtained values coincides with the available literature data in the case of x > 50 [41,42,43]. From the ratio of intensities of the second and fifth resonant lines of the Zeeman sextet to the first and sixth lines, respectively, we can determine the angle *ϑ* between the direction of gamma-quantum transit (axis of nanotubes) and the direction of the magnetic moment of the Fe atom:
(6)$$\frac{I_{2,5}}{I_{1,6}}=\frac{4\sin^{2}\vartheta }{3(1+\cos^{2}\vartheta )}$$


The average values of this angle as a function of Ni concentration are shown in Figure 7d. It can be seen that, both in the bcc region and in the fcc structure region, an increase in the concentration of Ni atoms led to a decrease in the mean angle *ϑ*. The random distribution of magnetic moments corresponded to *ϑ* = 54.7°. For nanotubes with a bcc structure, the average angle between the magnetic moment and the axis of the nanotubes decreased to ~40°, and with the fcc structure decreased from ~55° to ~46°. The change in the angle of the magnetic texture is due to the transition of the structure from the bcc to fcc with increasing nickel concentration in the structure. This transition is accompanied by the presence of two phases in the structure at a nickel concentration of 50–70% with a predominance of the fcc phase and an increase in nickel atoms in the lattice sites [47]. In this case, the presence of the contribution of the second phase lea to a partial disorder of the magnetic texture and a change in the angle. An increase in the nickel content in the structure led to the exclusion from the structure of the inclusions characteristic of the bcc structure and the ordering of the magnetic texture. Therefore, for studied nanotubes, a magnetic texture was observed along their axis.

### 3.2. Results of Model Interpretation of Mössbauer Spectra

Figure 8 shows the spectrum of Fe nanotubes and the results of its model interpretation. It can be seen that the main contribution was the sextet corresponding to nanotubes with the optimal values of the hyperfine parameters found with an accuracy determined by the standard deviations of statistical errors: *H_n_* = 327 kOe, *δ* = 0.003 ± 0.001 mm/s and *ε* = −0.001 ± 0.001 mm/s (the value of the chi-square functional χ^2^ = 1.2 ± 0.1). These values confirm that the nanotubes did consist of α-Fe.

The Mössbauer spectra of Fe_100-x_Ni_x_ nanotubes are shown in Figure 9 for x = 10, 20% and x = 30, 40%, respectively. The figures above the spectra depict bar graphs of positions of the resonance lines of each of the Zeeman sextets for Fe atoms with different nearest atomic environments. The arrangement of the bar charts from top to bottom corresponds to an increase in the number of Ni atoms in the nearest environment of the Fe atom. It is seen that for the Fe_100-x_Ni_x_ nanotubes with the bcc structure, the change in the hyperfine magnetic field ΔH_n_ caused by replacing the Fe atom with the Ni atom in the immediate environment of the iron atom was positive.

The spectra of nanotubes with fcc structure (with concentrations x = 50, 60, 70, 80, and 90%) are presented in Figure 10. Since in the case of an fcc structure, there can be from 0 to 12 nickel atoms in the nearest environment of the iron atom, the contribution from Fe_100-x_Ni_x_ nanotubes were processed by 13 Zeeman sextets. It can be seen (see the bar graphs above the spectra) that as the number of Ni atoms in the nearest environment of the Fe atom increased, the values of the hyperfine magnetic field for the corresponding sextets decreased, indicating a negative Δ*H_n_* value.

We note that the data for the hyperfine parameters of the spectra corresponding to the model interpretation, coincided with obtained values as a result of the restoration of distributions. The model interpretation of the Mössbauer spectra also made it possible to determine the average values of the angle between the axis of nanotubes and the magnetic moments of Fe atoms, depending on the concentration of Ni atoms, shown in Figure 11. In this case, the data obtained by two methods confirmed the presence of a magnetic texture for nanotubes and with bcc and with fcc structure.

From the concentration dependences presented above, it can be concluded that the results of the model decoding and restoration of distributions of the hyperfine parameters coincide. That confirms the assumptions made in the chosen model.

### 3.3. The Effect of the Replacement of Iron Atoms on Nickel Atoms in the Immediate Environment of Iron Atoms

Within the model, an assumption was made about the additivity of contributions to the hyperfine magnetic field, the shift of the Mössbauer line, and the quadrupole shift depending on the number m of Ni atoms in the nearest environment of Fe atom:
(7)$$H_{n}(m)=H_{n}(0)+m\Delta H_{n},$$
(8)δ(m)=δ(0)+mΔδ
(9)ε(m)=ε(0)+mΔε


This made it possible to obtain the values of changes in the hyperfine magnetic field Δ*H_n_*, the shift of the Mössbauer line Δ*δ*, and the quadrupole displacement Δ*ɛ* caused by the replacement of the Fe atom by the Ni atom in the nearest environment of the iron atom. The concentration dependence of the change in the hyperfine magnetic field is shown in Figure 12a. Replacing the Fe atom with the Ni atom in the nearest environment of the Fe atom in nanotubes with a bcc structure led to an increase in the hyperfine magnetic field on ^57^Fe nuclei by 6–9 kOe, and in tubes with a fcc structure to a decrease in the hyperfine magnetic field by 11 to 16 kOe. The dependences of the changes in the shift of the Mössbauer line Δ*δ* and the quadrupole shift Δ*ε* on the nickel concentration are shown in Figure 12b,c, respectively.

Replacing the Fe atom with the Ni atom in the nearest environment of the Fe atom in the bcc structure led to an increase in the shift of the Mössbauer line (from 0.02 mm/s to 0 mm/s, depending on the Ni concentration) and a decrease in the quadrupole shift by 0.02–0.03 mm/s Figure 11b,c). For the Fe_100-x_Ni_x_ nanotubes with the fcc structure, replacing the Fe atom by the Ni atom practically did not cause a shift in the Mössbauer line (|Δ*δ*| < 0.005 mm/s) and the quadrupole shifted by ~0.02 mm/s.

Knowing the values of changes in the hyperfine parameters caused by the replacement of the Fe atom by the Ni atom in the nearest environment of the iron atom, the ratios Δ*δ*/Δ*H_n_* (Figure 13a) and Δ*ε*/Δ*H_n_* (Figure 13b) were calculated.

It is seen that for an fcc structure, the ratio of the change in the shift of the Mössbauer line to the change in the hyperfine magnetic field caused by the replacement of the Fe atom by the Ni atom in the nearest environment of the iron atom was close to zero (Figure 12a), and the ratio of the change in quadrupole bias to the change in the hyperfine magnetic field was ~ −0.001 mm/s/кOe (Figure 12b). As a result of reconstructing the distributions of the hyperfine parameters of the Mössbauer spectrum, linear correlation coefficients $K_{\delta ,H_{n}}$ were obtained between the Mössbauer line shift *δ* and the hyperfine magnetic field *H_n_* (Figure 13c), as well as between the quadrupole shift *ε* and the hyperfine magnetic field *H_n_* (Figure 13d).

As can be seen, the linear correlation coefficient of the Mössbauer line *δ* shift and the hyperfine magnetic field *H*_n,_ when the position of the iron atom changes in the structure of nanotubes was approximately zero for nanotubes (Figure 12c). Also, a change in quadrupole displacement with a linear correlation coefficient $K_{\varepsilon ,H_{n}}$$K_{\varepsilon ,H_{n}}$ −(15 ± 5)·10^−4^ mm/s/кOe was observed correlated with the hyperfine magnetic field. Note that the concentration dependences of the linear correlation coefficients $K_{\delta ,H_{n}}$, and, $K_{\varepsilon ,H_{n}}$ and the ratios of changes in the hyperfine parameters, Δ*δ*/Δ*H_n_* and Δ*ε*/ΔH_n_, almost coincided. This means that distributions of the hyperfine parameters of the spectra were determined mainly by the degree of substitution of Fe atoms by Ni atoms in the nearest environment of the iron atom.

## 4. Conclusions

The main part of the obtained nanostructures was Fe_100-x_ Ni_x_ nanotubes with bcc structure for 0 ≤ x ≤ 40% and with fcc structure for 50 ≤ x ≤ 90%. The length, outside diameter and wall thickness of the nanotubes were 12 μm, 400 ± 10 nm and 120 ± 5 nm respectively. Extra subspectra of low intensity observed in experimental spectra presumably corresponds to the impurities of salts and magnetically ordered iron oxide compounds formed during synthesis. The concentration dependences of the hyperfine parameters of nanotube Mössbauer spectra are qualitatively consistent with the data for bulk polycrystalline samples. With Ni concentration increasing the average value of the hyperfine magnetic field increases from ~328 kOe to ~335 kOe for the bcc structure and drops to ~303 kOe in the transition to the fcc structure and then decreases to ~290 kOe at x = 90. The average value of the Mössbauer line shift increases from ~0 mm/s to ~0.045 mm/s for the bcc structure and decreases from ~0.04 mm/s to ~0.02 mm/s for the fcc structure. The average values of the quadrupole shift are close to zero (|ε| < 0.01 mm/s) but negative for the bcc structure and positive for the fcc structure. Magnetic texture is observed along axis of the studied nanotubes. The average value of the angle between the direction of the Fe atom magnetic moment and the nanotubes axis decreases with increasing of Ni concentration for nanotubes with bcc structure from ~50° to ~40° and with fcc structure from ~55° to ~46°. Replacing the Fe atom with Ni atom in the nearest environment of Fe atom within nanotubes with bcc structure lead to an increase in the hyperfine magnetic field on ^57^Fe nuclei by “6–9 kOe” and decrease of quadrupole shift by 0.02–0.03 mm/s, and in tubes with fcc structure—to a decrease in the hyperfine magnetic field by “11–16 kOe” and increase of quadrupole shift by ~0.02 mm/s. The changes of the quadrupole shift and hyperfine magnetic field from one to another position of Fe atom in the structure of nanotubes are linearly correlated with the coefficient −(15 ± 5)·10^−4^ mm/s/kOe. Such changes are caused by a variation in the number of Ni atoms in the nearest environment of an Fe atom.

## Figures and Tables

**Figure 1 nanomaterials-09-00757-f001:**
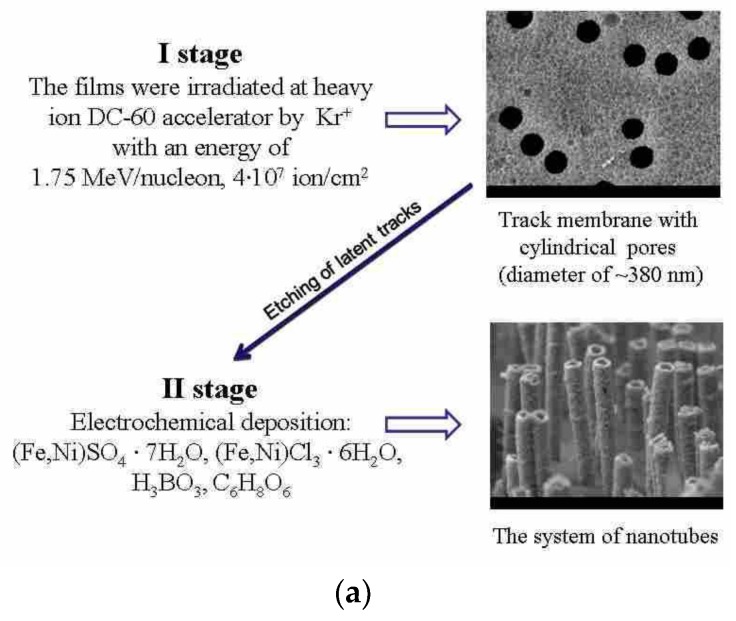
(**a**) Schematic representation of nanotubes synthesis; (**b**) SEM images of Fe_80_Ni_20_ nanotubes.

**Figure 2 nanomaterials-09-00757-f002:**
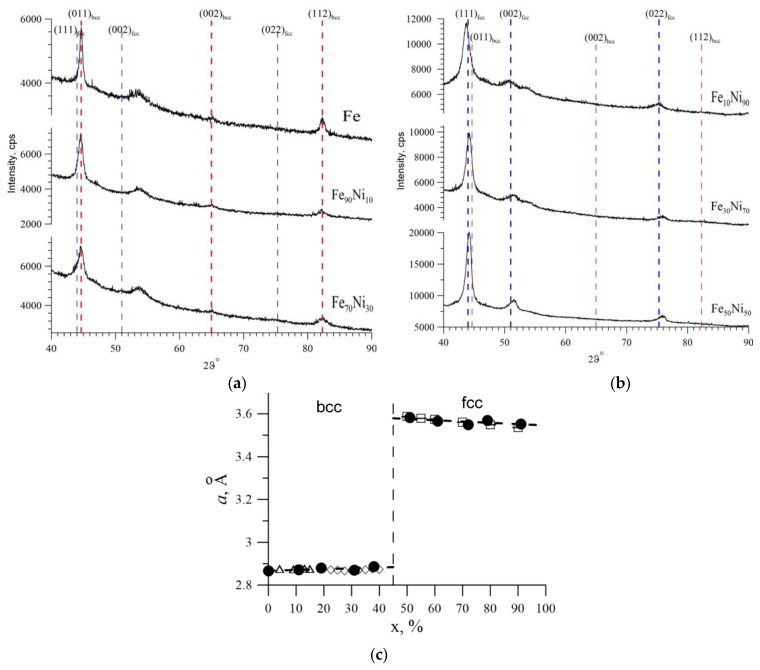
(**a**) Diffraction patterns of nanotubes with bcc structure. (**b**) Diffraction patterns of nanotubes with fcc structure. (**c**) The dependence of the unit cell parameter on the Ni concentration. •—experimental values obtained in this work, unpainted polygons—literature data [41,42,43].

**Figure 3 nanomaterials-09-00757-f003:**
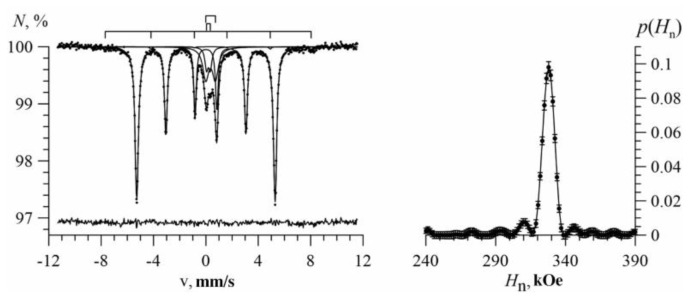
Mössbauer spectrum (left) and the result of reconstructing the distribution of the hyperfine magnetic field (right) of Fe nanotubes.

**Figure 4 nanomaterials-09-00757-f004:**
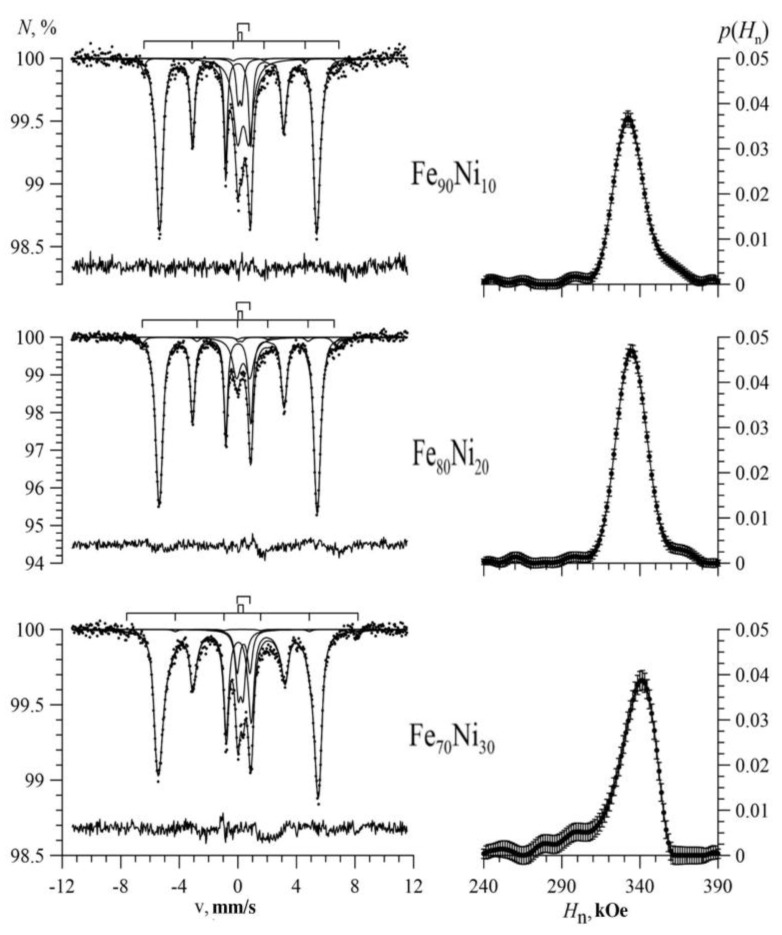
Mössbauer spectra (left) and the result of reconstructing the distribution of the hyperfine magnetic field (right) of Fe_100-x_Ni_x_ nanotubes at x = 10%; 20%; 30%.

**Figure 5 nanomaterials-09-00757-f005:**
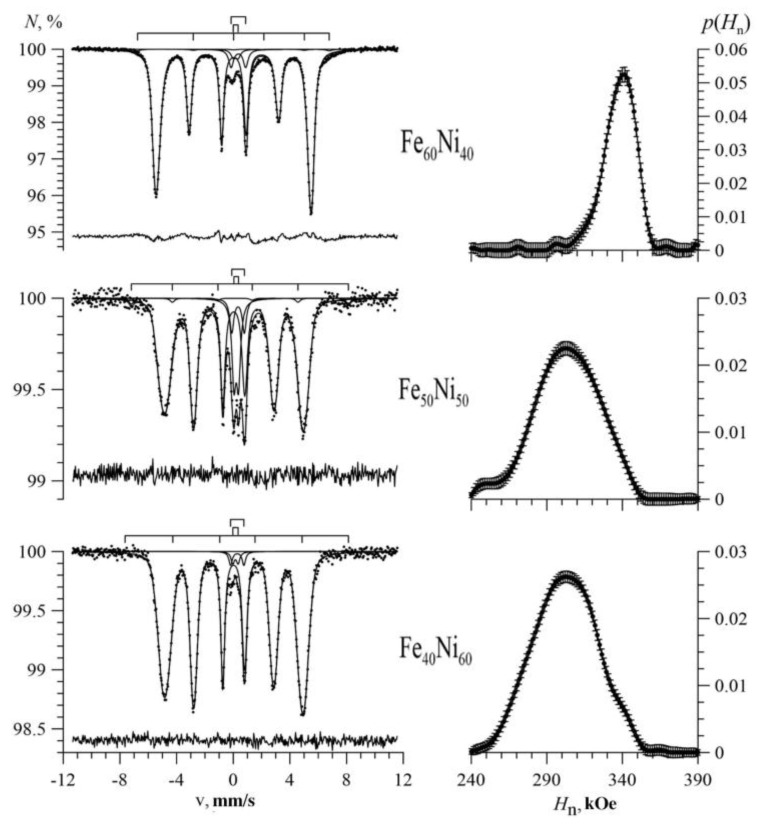
Mössbauer spectra (left) and the result of reconstructing the distribution of the hyperfine magnetic field (right) of Fe_100-x_Ni_x_ nanotubes at x = 40%; 50%; 60%.

**Figure 6 nanomaterials-09-00757-f006:**
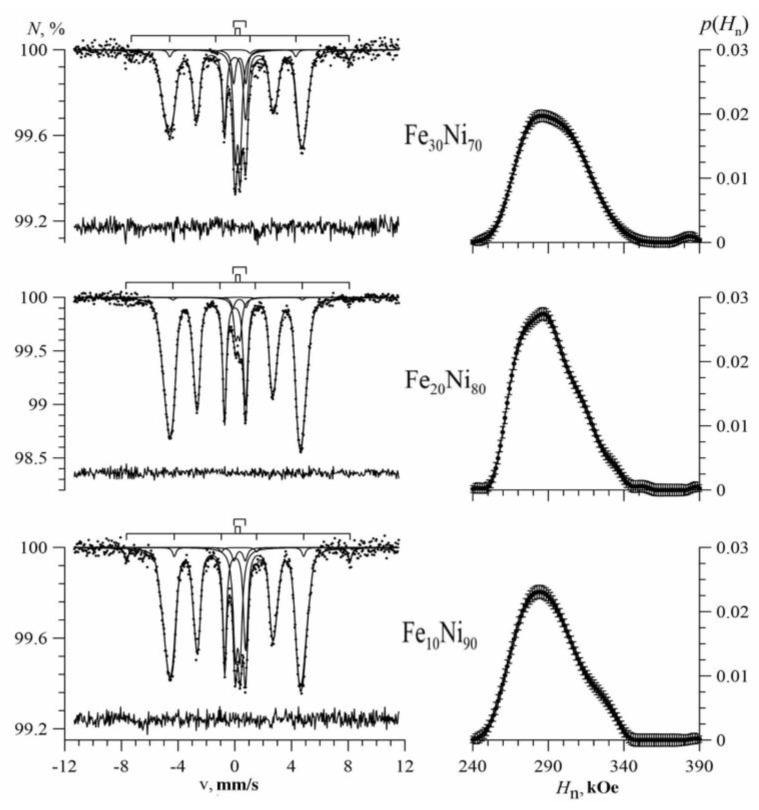
Mössbauer spectra (left) and the result of reconstructing the distribution of the hyperfine magnetic field (right) of Fe_100-x_Ni_x_ nanotubes at x = 70%; 80%; 90%.

**Figure 7 nanomaterials-09-00757-f007:**
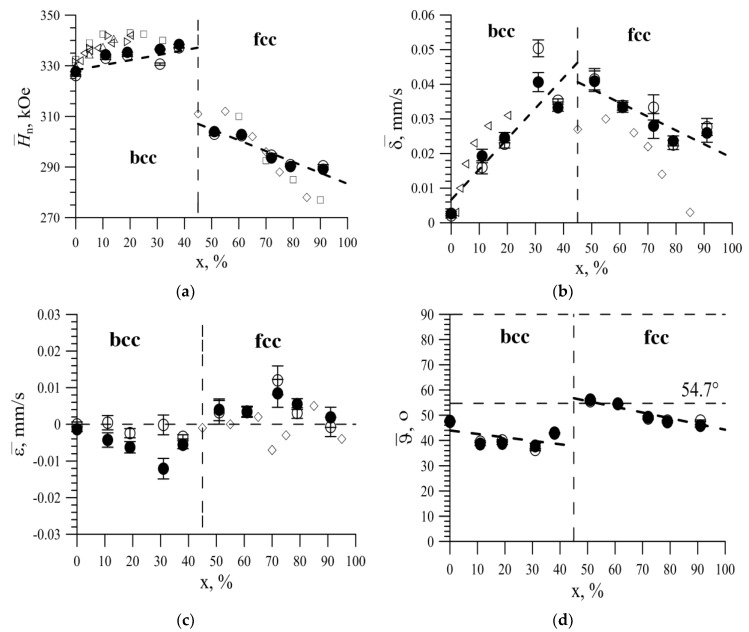
(**a**) The dependence of the average value of the hyperfine magnetic field *H_n_* on Ni concentration, obtained by restoring the distribution of the hyperfine magnetic field; (**b**) The dependence of the isomer shift of the Mössbauer line *δ* on Ni concentration, obtained by restoring the distributions of hyperfine parameters; (**c**) The concentration dependence of the mean value of the quadrupole shift, obtained as a result of the restoration of distributions of hyperfine parameters; (**d**) Concentration dependence of the average angle *ϑ* between the magnetic moment and the axis of the nanotubes, obtained by restoring the distribution of the hyperfine magnetic field. ●—the model fitting result; □, Δ, ◊—literary data [41,42,43]; ○—the reconstructed distribution of the magnetic field *p*(*H_n_*) result.

**Figure 8 nanomaterials-09-00757-f008:**
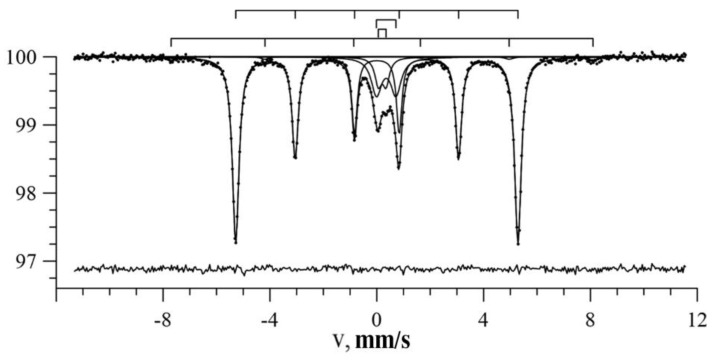
The result of the model interpretation of the Mössbauer spectrum of Fe nanotubes.

**Figure 9 nanomaterials-09-00757-f009:**
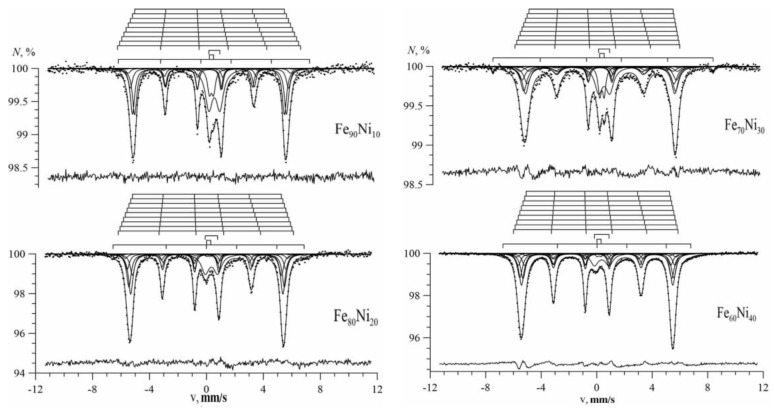
The result of the model decoding of the Mössbauer spectrum of Fe_100-x_Ni_x_ nanotubes.

**Figure 10 nanomaterials-09-00757-f010:**
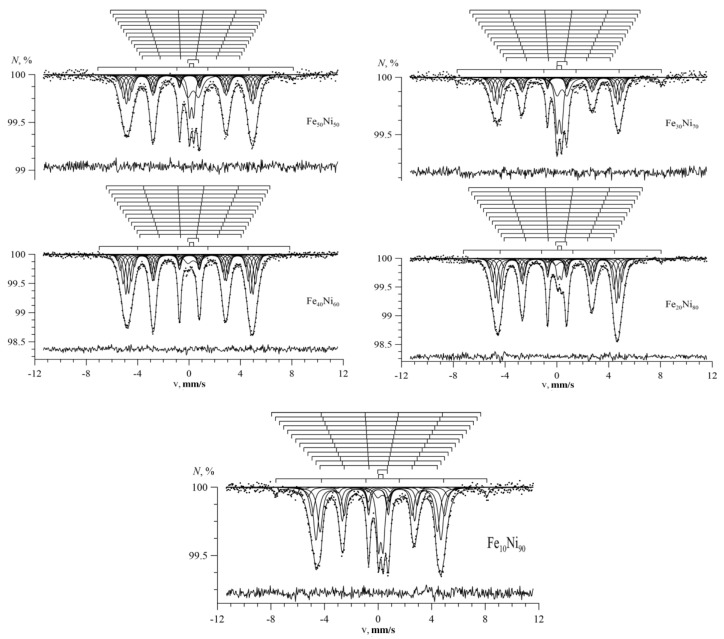
The result of the model decoding of the Mössbauer spectrum of Fe_100-x_Ni_x_ nanotubes.

**Figure 11 nanomaterials-09-00757-f011:**
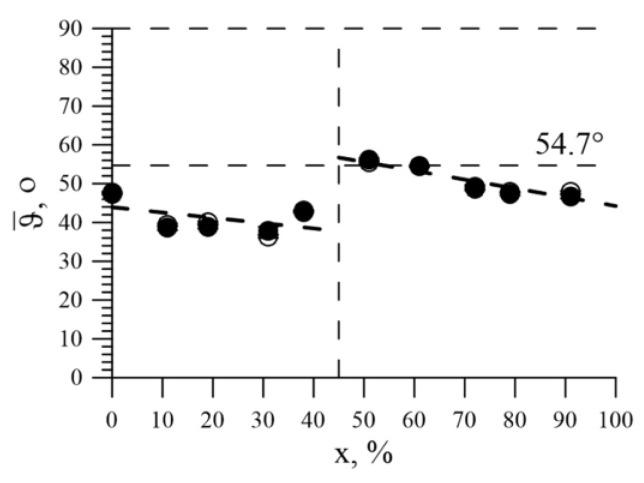
Concentration dependence of the average angle *ϑ* between the magnetic moments of Fe and the axis of nanotubes, obtained as a result of the restoration of distribution of hyperfine magnetic field (○) and the model interpretation.

**Figure 12 nanomaterials-09-00757-f012:**
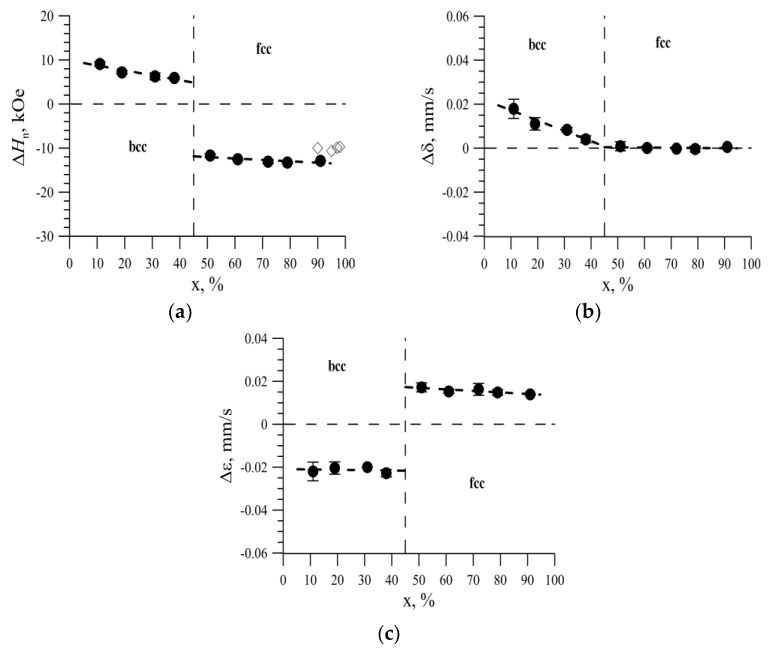
(**a**) The concentration dependence of the change in the hyperfine magnetic field Δ*H_n_*, caused by the replacement of the Fe atom by the Ni atom in the nearest environment of the iron atom; (**b**) The concentration dependence of the change in the shift of the Mössbauer line Δ*δ*, caused by the replacement of the Fe atom by the Ni atom in the nearest environment of the iron atom; (**c**) Concentration dependence of the quadrupole shift Δ*ɛ* change caused by the replacement of the Fe atom by the Ni atom in the nearest environment of the iron atom.

**Figure 13 nanomaterials-09-00757-f013:**
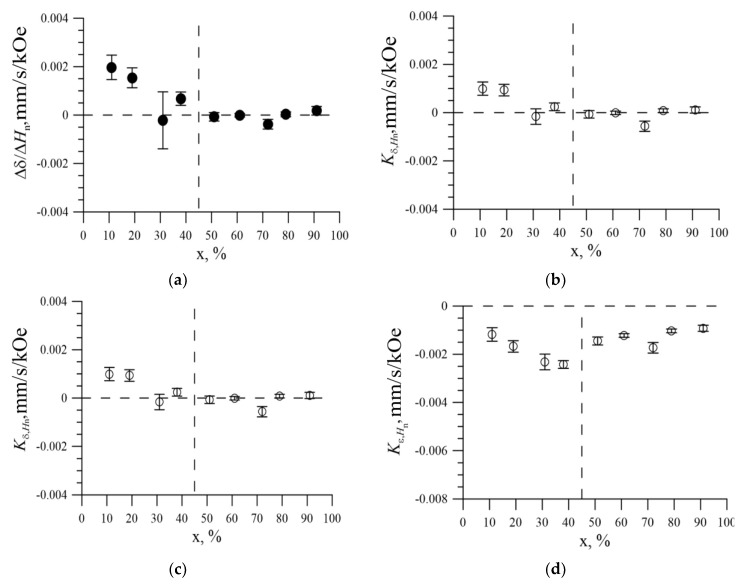
(**a**) The concentration dependence of the ratio of the change in the shift of the Mössbauer line to the change in the hyperfine magnetic field; (**b**) Concentration dependence of the ratio of the quadrupole shift change to the change in the hyperfine magnetic field; (**c**) The concentration dependence of the linear correlation coefficient of the shift of the Mössbauer line *δ* and the hyperfine magnetic field *H_n_* when the position of the iron atom changes in the structure of nanotubes; (**d**) Concentration dependence of the linear correlation coefficient of the quadrupole displacement of the line *ε* and the hyperfine magnetic field *H_n_* when the position of the iron atom changes in the structure of nanotubes.

**Table 1 nanomaterials-09-00757-t001:** The results of energy dispersive analysis of investigated nanostructures.

Sample	Atomic Ratio, %				
	Ni	Fe	Sample	Ni	Fe
Fe_100_	0	100	Fe_50_Ni_50_	51 ± 4	49 ± 3
Fe_90_Ni_10_	91 ± 4 *	9 ± 2	Fe_40_Ni_60_	38 ± 2	62 ± 4
Fe_80_Ni_20_	79 ± 4	21 ± 2	Fe_30_Ni_70_	31 ± 3	69 ± 3
Fe_70_Ni_30_	72 ± 4	28 ± 3	Fe_20_Ni_80_	19 ± 2	81 ± 4
Fe_60_Ni_40_	61 ± 3	39 ± 3	Fe_10_Ni_90_	11 ± 2	89 ± 4

* The accuracy of measurements was confirmed by measuring the spectra from different sites along the entire length of investigated samples in an amount of at least 10 spectra [44].

**Table 2 nanomaterials-09-00757-t002:** The data for hyperfine parameters.

Sample	PseudoVoight Sextet				PseudoVoight Doublet 1	PseudoVoight Doublet 2
	Intensity, %	I_2_/I_1_	Hyperfine Magnetic Field, kOe	Quadrupole Shift, mm/s	Intensity, %	Intensity, %
Fe_100_	79 ± 3	0.495	327.8 ± 0.2	−0.001 ± 0.001	15 ± 1	6 ± 1
Fe_90_Ni_10_	69 ± 2	0.337	332.5 ± 0.6	0.001 ± 0.003	25 ± 1	6 ± 1
Fe_80_Ni_20_	85 ± 3	0.346	333.4 ± 0.6	−0.001 ± 0.001	14 ± 1	1 ± 1
Fe_70_Ni_30_	82 ± 3	0.285	341.9 ± 0.6	−0.03 ± 0.01	8 ± 1	10 ± 1
Fe_60_Ni_40_	94 ± 2	0.405	342.3 ± 1.6	−0.02 ± 0.01	5 ± 1	1 ± 1
Fe_50_Ni_50_	85 ± 3	0.689	301.5 ± 0.5	0.01 ± 0.01	5 ± 1	10 ± 1
Fe_40_Ni_60_	97 ± 2	0.660	301.5 ± 0.3	0.01 ± 0.01	2 ± 1	1 ± 1
Fe_30_Ni_70_	76 ± 3	0.525	285.1 ± 0.7	0.03 ± 0.01	6 ± 1	18 ± 1
Fe_20_Ni_80_	93 ± 3	0.504	287.7 ± 3.1	0.01 ± 0.01	2 ± 1	5 ± 1
Fe_10_Ni_90_	83 ± 3	0.504	283.8 ± 0.5	0.01 ± 0.01	2 ± 1	15 ± 1

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
