# Peer review of "Study of Magnetic Properties of Fe_100-x_Ni_x_ Nanostructures Using the Mössbauer Spectroscopy Method"

_nanomaterials, 2019, doi:10.3390/nano9050757_

Round 1
Reviewer 1 Report
The manuscript presents a comprehensive study of Fe100-xNix nanostructures synthesized in polymer ion-track membranes. The set of data is sufficient and the analysis and interpretation is accurate. However, the technical quality of presentation should be improved, namely:
- the font size in the Moessbauer spectra should be enlarged and the resolution of the images should be improved (Figs 3-6, 8, 9).
- there are quite frequent typos and strange characters, especially in formulas (p. 12, 13), Mossbauer vs. Moessbauer etc.
- use of upper/lower index should be corrected, i.e. 57Fe vs 57Fe
- the details on data treatment and probably some of the spectra should be shifted to SI (section 3.2)
Author Response
1. It has been corrected.
2. It has been corrected.
3. It has been corrected.
4. It has been corrected. The description of method is shifted.

Reviewer 2 Report
Please see the attached referee report.

Author Response
These conditions allow obtaining cylindrical pores with a diameter of 380 ± 10 nm (according to SEM images, Fig. 1b).
Figure 2a shows, as an example, the diffractograms of some of the studied samples in which the nickel concentration is lower than the iron concentration. It can be seen that the intensity peaks in these diffractograms are at an angle of 2ϑ, equal to 44.6⁰, 65.0⁰ and 82.3⁰, with the corresponding Miller indices: (011), (002) and (112), which confirms the body-centered cubic structure (bcc). Figure 2b shows the diffraction patterns for samples in which the nickel content exceeds the iron content. Similarly, from the obtained data on diffraction angles of 2ϑ, the values of which were 44.0⁰, 51.0⁰ and 75.3⁰ with the corresponding Miller indices (111), (002) and (022), the crystal structure was determined. For these samples, a face-centered cubic structure (fcc) is observed. The diffraction line at an angle of ~ 53.7°, observed for all samples, corresponds to the PET pattern template.
Figure 3 shows the Mossbauer spectrum of samples for Fe nanotubes containing no nickel.
All data obtained from the analysis of the Mossbauer spectra are presented in the form of graphs of dependencies on the nickel concentration in the structure in Figures 7, 11, and 12.
The average values of this angle as a function of Ni concentration are shown in Figure 7d. It can be seen that, both in the bcc region and in the fcc structure region, an increase in the concentration of Ni atoms leads to a decrease in the mean angle ϑ. The random distribution of magnetic moments corresponds to ϑ = 54.7⁰. For nanotubes with a bcc structure, the average angle between the magnetic moment and the axis of the nanotubes decreases to ~ 40º, and with the fcc structure decreases from ~ 55º to ~ 46º. The change in the angle of the magnetic texture is due to the transition of the structure from the bcc to fcc with increasing nickel concentration in the structure. This transition is accompanied by the presence of two phases in the structure at a nickel concentration of 50–70% with a predominance of the fcc phase and an increase in nickel atoms in the lattice sites [47]. In this case, the presence of the contribution of the second phase leads to a partial disorder of the magnetic texture and a change in the angle. An increase in the nickel content in the structure leads to the exclusion from the structure of the inclusions characteristic of the bcc structure and the ordering of the magnetic texture. Therefore, for studied nanotubes, a magnetic texture is observed along their axis.
It has been corrected.
It has been corrected.

Round 2
Reviewer 2 Report
Please see the attached referee report.

Author Response
Correct
Correct
Correct
Correct
Correct

Round 3
Reviewer 2 Report
Please see the attached referee report.

Author Response
Correct
Correct
Correct
Correct
Correct
Correct
Correct
